# MetaSDF: Meta-learning Signed Distance Functions

**Vincent Sitzmann**[*]
Stanford University
sitzmann@cs.stanford.edu

**Eric R. Chan**[*]
Stanford University
erchan@stanford.edu

**Richard Tucker**
Google Research
richardt@google.com

**Noah Snavely**
Google Research
snavely@google.com

**Gordon Wetzstein**
Stanford University
gordon.wetzstein@stanford.edu

vsitzmann.github.io/metasdf/

## Abstract

Neural implicit shape representations are an emerging paradigm that offers many potential benefits over conventional discrete representations, including memory efficiency at a high spatial resolution. Generalizing across shapes with such neural implicit representations amounts to learning priors over the respective function space and enables geometry reconstruction from partial or noisy observations. Existing generalization methods rely on conditioning a neural network on a low-dimensional latent code that is either regressed by an encoder or jointly optimized in the auto-decoder framework. Here, we formalize learning of a shape space as a meta-learning problem and leverage gradient-based meta-learning algorithms to solve this task. We demonstrate that this approach performs on par with auto-decoder based approaches while being an order of magnitude faster at test-time inference. We further demonstrate that the proposed gradient-based method outperforms encoder-decoder based methods that leverage pooling-based set encoders.

## 1 Introduction

Humans possess an impressive intuition for 3D shapes; given partial observations of an object we can easily imagine the shape of the complete object. Computer vision and machine learning researchers have long sought to reproduce this ability with algorithms. An emerging class of neural implicit shape representations, for example using signed distance functions parameterized by neural networks, promises to achieve these abilities by learning priors over neural implicit shape spaces [1, 2]. In such methods, each shape is represented by a function $\Phi$, e.g., a signed distance function. Generalizing across a set of shapes thus amounts to learning a prior over the space of these functions $\Phi$. Two questions arise: (1) How do we parameterize functions $\Phi$, and (2) how do we infer the parameters of such a $\Phi$ given a set of (partial) observations?

Existing methods assume that the space of functions $\Phi$ is low-dimensional and represent each shape as a latent code, which parameterizes the full function $\Phi$ via concatenation-based conditioning or hypernetworks. These latent codes are either directly inferred by a 2D or 3D convolutional encoder, taking either a single image or a classic volumetric representation as input, or via the auto-decoder framework, where separate latent codes per training sample are treated as free variables at training time. Convolutional encoders, while fast, require observations on a regular grid, and are not equivariant to 3D transformations [3]. They further do not offer a straightforward way to accumulate information from a variable number of observations, usually resorting to permutation-invariant pooling of per-observation latent codes [4]. Recently proposed pooling-based set encoders may encode sets of

---

[*]These authors contributed equally to this work.

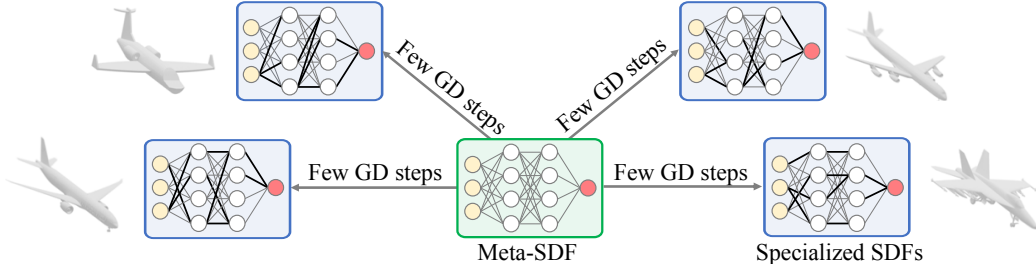

Figure 1: We propose to leverage gradient-based meta-learning to learn the initialization of an implicit neural representation such that it can be adapted to any specific instance in few gradient descent steps.

variable cardinality [5, 6], but have been found to underfit the context. This is corroborated by theoretical evidence, showing that to guarantee universality of the embedded function, the embedding requires a dimensionality of at least the number of context points [7]. In practice, most approaches thus leverage the auto-decoder framework for generalization, which is agnostic to the number of observations and does not require observations on a regular grid. This has yielded impressive results on few-shot reconstruction of geometry, appearance and semantic properties [8, 9].

To infer the parameters of a single $\Phi$ from a set of observations at test time, encoder-based methods only require a forward pass. In contrast, the auto-decoder framework does not learn to infer an embedding from observations, and instead requires solving an optimization problem to find a low-dimensional latent embedding at test time, which may take several seconds even for simple scenes, such as single 3D objects from the ShapeNet dataset.

In this work, we identify a key connection between learning of neural implicit function spaces and meta-learning. We then propose to leverage recently proposed gradient-based meta-learning algorithms for the learning of shape spaces, with benefits over both encoder and auto-decoder based approaches. Specifically, the proposed approach performs on par with auto-decoder based models while being an order of magnitude faster at inference time, does not require observations on a regular grid, naturally interfaces with a variable 'number of observations, outperforms pooling-based set-encoder approaches, and does not require the assumption of a low-dimensional latent space. We note that we do not propose any improvements to gradient-based meta-learning algorithms in general.

## 2 Related Work

**Neural Implicit Scene Representations.** Implicit parameterizations of scenes are an emerging topic of interest in the machine learning community. Parameterized as multilayer perceptrons, these continuous representations have applications in modeling shape parts [10, 11], objects [1, 12–14], or scenes [8, 15, 16]. These and related neural implicits are typically trained from 3D data [1, 2, 12–19], but more recent work has also shown how 2D image data can be directly used to supervise the training procedure [8, 20–22], leveraging differentiable neural rendering [23]. Earlier work on compositional pattern–producing networks explored similar strategies to parameterize 2D images [24, 25].

**Learning shape spaces.** A large body of prior work has explored learning priors over the parameters of classic geometry representations, such as meshes, voxel grids, or point clouds [5, 26–28]. Learning a prior over neural implicit representations of geometry, however, requires learning a prior over a space of *functions*. To this end, existing work assumes a low-dimensional latent shape space, and leverages auto-decoders [1, 8] or convolutional encoders [2, 19, 21] to regress a latent embedding. While generally, a single latent code represents the full object, making them a function of spatial coordinates enables capturing of local features [18, 19]. This embedding is then decoded into a function either via concatenation-based conditioning [1, 2, 19] or via hypernetworks [8, 29]. In this work, we instead propose a meta-learning-based approach to learning a shape of signed distance functions.

**Meta-Learning.** Meta-learning is usually formalized in the context of few-shot learning. Here, the goal is to train a learner that can quickly adapt to new, unseen tasks given only few training examples, often referred to as context observations. One class of meta-learners proposes to learn an optimizer or update rule [30–32]. Conditional and attentive neural processes [4, 6, 33] instead encode

context observations into a low-dimensional embedding via a permutation-invariant set encoder, and a decoder network is conditioned on the resulting latent embedding. A recent class of algorithms proposes to learn the initialization of a neural network, which is then specialized to a new task via few steps of gradient descent [34, 35]. This class of algorithms has recently seen considerable interest, resulting in a wide variety of extensions and improvements. Rusu et al. [36] blend feed-forward and gradient-descent based specialization via optimization in a latent space. Rajeswaran et al. [37] propose to obtain gradients via an implicit method instead of backpropagating through unrolled iterations of the inner loop, leading to significant memory savings. In this work, we leverage this class of gradient-descent based meta learners to tackle the learning of a shape space. We refer the reader to a recent survey paper for an exhaustive overview [38].

## 3 Meta-learning Signed Distance Functions

We are interested in learning a prior over implicit shape representations. As in [1], we consider a dataset $\mathcal{D}$ of $N$ shapes. Each shape is itself represented by a set of points $X_i$ consisting of $K$ point samples from its ground-truth signed distance function (SDF) $\text{SDF}_i$:

$$\mathcal{D} = \{X_i\}_{i=1}^N, \quad X_i = \{(\mathbf{x}_j, s_j) : s_j = \text{SDF}_i(\mathbf{x}_j)\}_{j=1}^K \tag{1}$$

Where $\mathbf{x}_j$ are spatial coordinates, and $s_j$ are the signed distances at these spatial coordinates. We aim to represent a shape by directly approximating its SDF with a neural network $\Phi_i$ [1], which represents the surface of the shape implicitly as its zero-level set $L_0(\Phi_i)$:

$$L_0(\Phi_i) = \{x \in \mathbb{R}^3 | \Phi_i(x) = 0\}, \quad \Phi_i : \mathbb{R}^3 \mapsto \mathbb{R} \tag{2}$$

We may alternatively choose to approximate the binary 'occupancy' of points rather than their signed distances; the surface would then be represented by the binary classifier's decision boundary [2].

Generalizing across a set of shapes thus amounts to learning a prior over the space of functions $\mathcal{X}$ where $\Phi_i \in \mathcal{X}$. Two questions arise: (1) How do we parameterize a single element $\Phi_i$ of the set $\mathcal{X}$, and (2) how do we infer the parameters of such a $\Phi_i$ given a set of (partial) observations $X_i$?

**Parameterizing $\Phi_i$: Conditioning via concatenation and hypernetworks.** Existing approaches to generalizing over shape spaces rely on decoding latent embeddings of shapes. In conditioning via concatenation, the target coordinates $\mathbf{x}$ are concatenated with the latent shape embedding $\mathbf{z}_i$ and are fed to a feedforward neural network whose parameters are shared across all shape instances. The latent code conditions the output of the shared network and allows a single shared network to represent multiple shape instances [1, 2].

In an alternative formulation, we can instead use a hypernetwork, which takes as input the latent code $\mathbf{z}_i$ and generates the parameters of a shape-representing MLP, $\Phi_i$. $\Phi_i$ can then be sampled at coordinates $\mathbf{x}$ to produce output signed distances. As we demonstrate in the supplement, conditioning via concatenation is a special case of a hypernetwork [29], where the hypernetwork is parameterized as a single linear layer that predicts only the biases of the network $\Phi_i$. Indeed, recent related work has demonstrated that hypernetworks mapping a latent code $\mathbf{z}_i$ to *all* parameters of the MLP $\Phi_i$ can similarly be used to learn a space of shapes [8].

**Inferring parameters of $\Phi_i$ from observations: Encoders and Auto-decoders.** Existing approaches to inferring latent shape embeddings $\mathbf{z}_i$ rely on encoders or auto-decoders. The former rely on an encoder, such as a convolutional or set encoder, to generate a latent embedding that is then decoded into an implicit function using conditioning via concatenation or hypernetworks.

In auto-decoder (i.e., decoder-only) architectures, the latent embeddings are instead treated as learned parameters rather than inferred from observations at training time. At training time, the auto-decoder framework enables straightforward generalization across a set of shapes. At test time, we freeze the weights of the model and perform a search to find a latent code $\hat{\mathbf{z}}$ that is compliant with a set of context observations $\hat{X}$. A major weakness of the auto-decoder framework is that solving this optimization problem is slow, taking several seconds per object.

### 3.1 Shape generalization as meta-learning

Meta-learning aims to learn a model that can be quickly adapted to new tasks, possibly with limited training data. When adapting to a new task, the model is given 'context' observations, with which

**Algorithm 1** MetaSDF: Gradient-based meta-learning of shape spaces

---

**Precondition:** Distribution $\mathcal{D}$ over SDF samples, outer learning rate $\beta$, number of inner-loop steps $k$

1: Initialize inner, per-parameter learning rates $\alpha$, meta-parameters $\theta$
2: **while** not done **do**
3:     Sample batch of shape datasets $X_i \sim \mathcal{D}$
4:     **for all** $X_i$ **do**
5:         Split $X_i$ into $X_i^{train}$, $X_i^{test}$
6:         Initialize $\phi_i^0 = \theta$, $\mathcal{L}_{train} = 0$
7:         **for** $j = 0$ **to** $k$ **do**
8:             $\mathcal{L}_{train} \leftarrow \frac{1}{|X_i^{train}|} \sum\limits_{(\mathbf{x},s) \in X_i^{train}} \ell_1 \left( \Phi(\mathbf{x}; \phi_i^j), s \right)$

9:             $\phi_i^{j+1} \leftarrow \phi_i^j - \alpha \odot \nabla_{\phi_i^j} \mathcal{L}_{train}$

10:        $\mathcal{L}_{test} \leftarrow \mathcal{L}_{test} + \frac{1}{|X_i^{test}|} \sum\limits_{(\mathbf{x},s) \in X_i^{test}} \ell_1 \left( \Phi(\mathbf{x}; \phi_i^k), s \right)$

11:    $\theta, \alpha \leftarrow (\theta, \alpha) - \beta \nabla_{(\theta, \alpha)} \mathcal{L}_{test}$
     **return** $\theta, \alpha$

---

it can modify itself, e.g. through gradient descent. The adapted model can then be used to make predictions on unseen 'target' observations. Formally, supervised meta-learning assumes a distribution over tasks $\mathcal{T} \sim p(\mathcal{T})$, where each task is of the form $\mathcal{T} = (\mathcal{L}, \{(\mathbf{x}_l, \mathbf{y}_l)\}_{l \in \mathbf{C}}, \{(\mathbf{x}_j, \mathbf{y}_j)\}_{j \in \mathbf{T}})$, with loss function $\mathcal{L}$, model inputs $\mathbf{x}$, model outputs $\mathbf{y}$, the set of all indices belonging to a context set $\mathbf{C}$, and the set of all indices belonging to a target set $\mathbf{T}$. At training time, a batch of tasks $\mathcal{T}_i$ is drawn, with each task split into 'context' and 'target' observations. The model adapts itself using the 'context' observations and makes predictions on the 'target' observations. The model parameters are optimized to minimize the losses $\mathcal{L}$ on all target observations in the training set, over all tasks.

Our key idea is to view the learning of a shape space as a meta-learning problem. In this framework, a task $\mathcal{T}_i$ represents the problem of finding the signed distance function $\Phi_i$ of a shape. Simply put, by giving a model a limited number of 'context' observations, each of which consists of a world coordinate location $\mathbf{x}$ and a ground truth signed-distance $s$, we aim to quickly specialize it to approximating the underlying SDF$_i$. Context and target observations are samples from an object-specific dataset $X_i$: $\mathcal{T}_i = (\{(\mathbf{x}_l, s_l)\}_{l \in \mathbf{C}}, \{(\mathbf{x}_j, s_j)\}_{j \in \mathbf{T}})$. Because each task consists of fitting a signed distance function, we can choose a global loss, such as $\ell_1$.

Note that the auto-decoder framework could be viewed as a meta-learning algorithm as well, though it is not generally discussed as such. In this view, the auto-decoder framework is an outlier, as it does not perform model specialization in the forward pass. Instead, specialization requires stochastic gradient descent until convergence, both at training and at test time.

**Learning a shape space with gradient-based meta-learning.** We propose to leverage the family of MAML-like algorithms [34]. We consequently view an instance-specific SDF$_i$, approximated by $\Phi_i$, as the specialization of an underlying meta-network with parameters $\theta$. In the forward pass, we sample a batch of shape datasets $X_i$, and split each dataset into a training set $X_i^{train}$ and test set $X_i^{test}$. We then perform $k$ gradient descent steps according to the following update rule:

$$\phi_i^{j+1} = \phi_i^j - \lambda \nabla \sum_{(\mathbf{x},s) \in X_i^{train}} \mathcal{L}(\Phi(\mathbf{x}; \phi_i^j), s), \quad \phi_i^0 = \theta \tag{3}$$

where $\phi_i^j$ are the parameters for shape $i$ at inner-loop step $j$, and $\Phi(\cdot\,; \phi_i^j)$ indicates that the network $\Phi$ is evaluated with these parameters. The final specialized parameters $\phi_i^k$ are now used to make predictions on the test set, $X_i^{test}$, and to compute an outer-loop loss $\mathcal{L}_{test}$. Finally, we backpropagate the loss $\mathcal{L}_{test}$ through the inner-loop update steps to the parameters $\theta$. We further found that the proposed method benefited from the added flexibility of per-parameter learning rates, as proposed by Li et al. [39]. The full algorithm is formalized in Algorithm 1.

This formulation has several advantages over the auto-decoder framework. First, as we demonstrate in Section 4, inference at test time is an order of magnitude faster, while performing on par or slightly better both qualitatively and quantitatively. Further, as MAML optimizes the inference algorithm as well, it can be trained to infer the specialized network from different kinds of context observations. We will demonstrate that this enables, for instance, reconstruction of an SDF given only points on the zero-level set, while the auto-decoder framework requires heuristics in the form of surrogate

| | GT | HyperNet | Concat. | Cond. NP | MetaSDF |
|---|---|---|---|---|---|

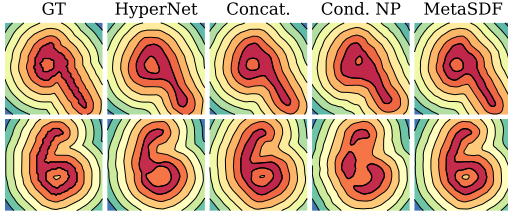

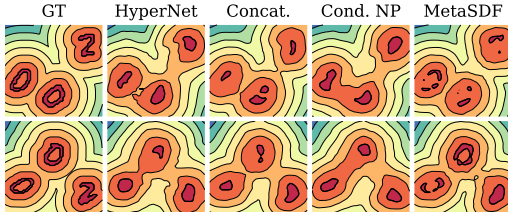

Figure 3: Reconstruction of SDFs of two test digits from classes unseen at training time.

| | GT | HyperNet | Concat. | Cond. NP | MetaSDF |
|---|---|---|---|---|---|

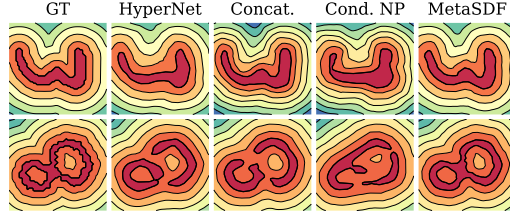

Figure 4: Reconstruction of SDFs of two rotated test digits.

| | GT | HyperNet | Concat. | Cond. NP | MetaSDF |
|---|---|---|---|---|---|

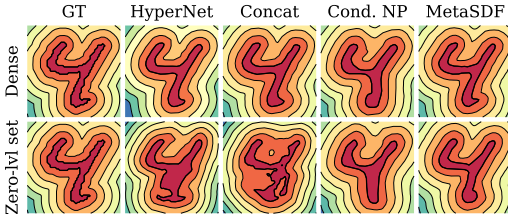

Figure 5: Reconstruction of SDFs of two examples with three digits.

| | Unseen Digit | Rotation | Composit. |
|---|---|---|---|
| HyperNet | 0.144 | 0.189 | 0.284 |
| Concat | 0.062 | 0.151 | 0.137 |
| Cond. NP | 0.253 | 0.316 | 0.504 |
| MetaSDF | **0.023** | **0.032** | **0.059** |

Table 1: $\ell_1$-error for reconstruction of out-of-distribution samples. All results are $\times 10^{-3}$.

| | GT | HyperNet | Concat | Cond. NP | MetaSDF |
|---|---|---|---|---|---|

Dense

Zero-lvl set

| | Dense | Zero-level set |
|---|---|---|
| HyperNet | 0.044 | 1.562 |
| Concat | 0.050 | 1.813 |
| Cond. NP | 0.108 | 0.156 |
| MetaSDF | **0.015** | **0.057** |

Figure 6: Qualitative and quantitative comparison of performance of different generalization strategies in inferring the signed distance function of an MNIST digit either from dense observations (top row), or only points on the zero-level set (bottom row). All quantitative results are $\ell_1$-error $\times 10^{-3}$.

losses to achieve this goal. We empirically show that we outperform pooling-based set-encoder based methods, consistent with recent work that found these encoders to underfit the context [33]. Finally, both auto-decoder and auto-encoder-based methods assume a low-dimensional latent space, while MAML naturally optimizes in the high-dimensional space of all parameters $\theta$ of the meta-network.

## 4 Analysis

In this section, we first apply the proposed MetaSDF approach to the learning of 2D SDFs extracted from MNIST digits, and subsequently, on 3D shapes from the ShapeNet dataset. All code and datasets will be made publicly available.

### 4.1 Meta-learning 2D Signed Distance Functions

We study properties of different generalization methods on 2D signed distance functions (SDFs) extracted from the MNIST dataset. From every MNIST digit, we extract a 2D SDF via a distance transform, such that the contour of the digit is the zero-level set of the corresponding SDF, see Fig. 2. Following [1], we directly fit the SDF of the MNIST digit via a fully connected neural network. We benchmark three alternative generalization approaches. First, two auto-decoder based approaches,

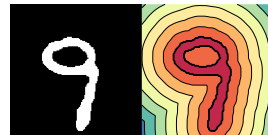

Figure 2: MNIST digit with extracted SDF, bold zero-level set.

where the latent code is decoded into a function $\Phi$ either using conditioning via concatenation as in [1] or via a fully connected hypernetwork as in [8]. Second, we compare to a conditional neural process (CNP) [6], representative of permutation-invariant set-encoders.

**Implementation.** All models are implemented as fully connected ReLU-MLPs with 256 hidden units and no normalization layers. $\Phi$ is implemented with four layers. The set encoder of CNPs

similarly uses four layers. Hypernetworks are implemented with three layers as in [8]. The proposed approach performs 5 inner-loop update steps, where we initialize $\alpha$ as $1 \times 10^{-1}$. All models are optimized using the ADAM optimizer [40] with a learning rate of $1 \times 10^{-4}$.

**Inference with partial observations.** We demonstrate that we may train the MetaSDF to infer continuous SDFs either from dense ground-truth samples, or from samples from *only the zero-level set*. This is noteworthy, as inferring an SDF from only the zero-level set points amounts to solving a boundary value problem defined by a particular *Eikonal equation*, see [14]. In DeepSDF [1], the authors also demonstrate test-time reconstruction from zero-level set points, but require augmenting the loss with heuristics to ensure the problem is well posed. Gropp et al. [14] instead propose to explicitly account for the Eikonal constraint in the loss. We train all models on SDFs of the full MNIST training set, providing supervision via a regular grid of $64 \times 64$ ground-truth SDF samples. For CNPs and the proposed approach, we train two models each, conditioned on either (1) the same $64 \times 64$ ground-truth SDF samples or (2) a set of $512$ points sampled from the zero-level set. We then test all models to reconstruct SDFs from the unseen MNIST test set from these two different kinds of context. Fig. 6 shows a qualitative and quantitative comparison of results. All approaches qualitatively succeed in reconstructing SDFs with a dense context set. Quantitatively, MetaSDFs perform on par with auto-decoder based methods for conditioning on dense SDF values. Both methods outperform CNPs by an order of magnitude. This is consistent with prior work showing that pooling-based set encoders tend to underfit the context set. When conditioned on the zero-level set only, auto-decoder based methods fail to reconstruct test SDFs. In contrast, both CNPs and MetaSDFs succeed in reconstructing the SDF, with the proposed approach again significantly outperforming CNPs.

**Inference speed.** We compare the time required to fit a single unseen 2D MNIST SDF for both hypernetworks and the concatenation-based approach. Even for this simple example, both methods require on the order of $4$ seconds to converge on the latent code of an unseen SDF at test time. In contrast, the proposed meta-learning based approach infers functional representations in 5 gradient descent steps, or in about $50$ ms.

**Out-of-distribution generalization.** We investigate the capability of different optimization methods to reconstruct out-of-distribution test samples. We consider three modes of out-of-distribution generalization: First, generalization to MNIST digits unobserved at training time. Here, we train on digits 0–5, holding out 6–9. Second, generalization to randomly rotated MNIST SDFs, where we train on the full MNIST training set. And last, generalization to signed distance functions extracted from the triple-MNIST dataset [41], a dataset of compositions of three scaled-down digits into a single image. In this last experiment, we train on a dataset comprised of both the full MNIST training set, as well as the double-MNIST dataset, testing on the triple-MNIST dataset. The double-MNIST dataset contains digits of the same size as the target triple-MNIST dataset. Figures 3, 4, and 5 show qualitative results, while Table 1 reports quan-

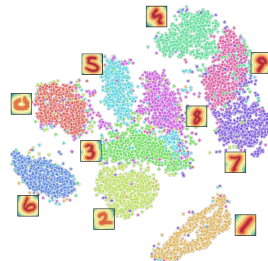

Figure 7: T-SNE embedding of *parameters* of specialized networks.

titative performance. The proposed MetaSDF approach far outperforms all alternative approaches in this task. Qualitatively, it succeeds in generating both unseen digits and rotated digits, although no method succeeds to accurately reconstruct the zero-level set in this compositionality experiment. This suggests that the proposed approach is more flexible to out-of-distribution samples.

**Interpretation as representation learning.** It has previously been observed that learning implicit representations can be seen as a form of representation learning [9]. Here, we corroborate this observation, and demonstrate that weights of specialized SDF networks found with the proposed meta-learning approach encode information about the digit class, performing unsupervised classification. Fig. 7 shows T-SNE [42] embeddings of the parameters of $10,000$ test-set MNIST SDFs. While classes are not perfectly linearly separable, it is apparent that the parameters of the SDF-encoding neural network $\Phi$ carry information about the class of the MNIST digit.

### 4.2   Meta-Learning a 3D Shape Space

We now demonstrate that the proposed approach scales to three dimensions, where results on 3D shape representation are consistent with the 2D MNIST results above.

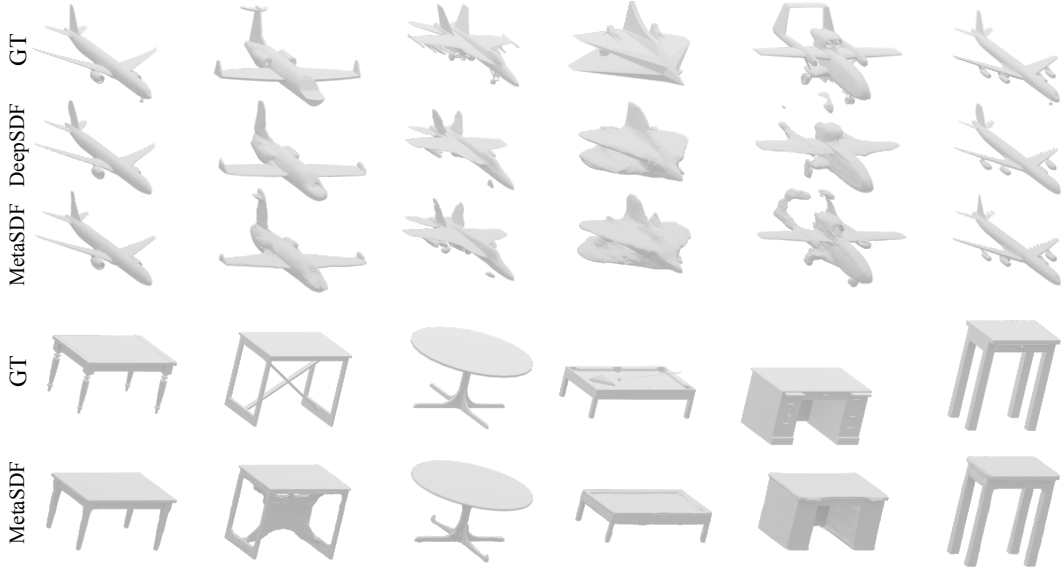

Figure 8: Top row: comparison of reconstructed shapes of DeepSDF and MetaSDF on the ShapeNet planes class. Both models receive dense signed distance values at training and test time. The two methods perform approximately on par, with the proposed method performing slightly better on out-of-distribution samples while speeding up inference by one order of magnitude. Bottom row: Additional reconstructions of MetaSDF from the ShapeNet tables class.

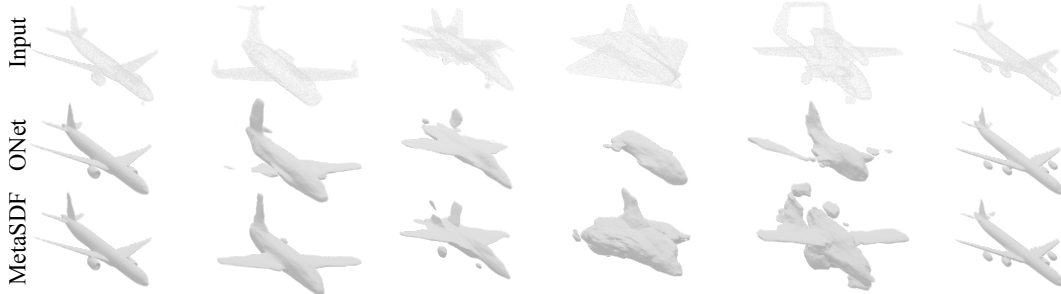

Figure 9: Comparison of reconstructed shapes of proposed method and a PointNet-encoder based method as proposed in [2]. Both models are conditioned only on zero-level set points at training and test time. The proposed method generally outperforms the PointNet-based approach, with strongest gains in uncommon airplanes.

We first benchmark the proposed approach in the case of a densely sampled SDF. To this end, we follow the experimental setup of Park et al. [1], and benchmark with their DeepSDF model. DeepSDF combines the auto-decoder framework with conditioning via concatenation to learn a prior over 3D shapes in the ShapeNet [43] dataset. The dataset contains ground-truth samples from the analytically computed SDFs of the respective mesh. The application of the proposed framework to this dataset follows the 2D examples in Sec. 4.1, with two minor differences. Park et al. [1] propose to clamp the $\ell_1$ loss to increase accuracy of the SDF close to the surface. We empirically found that clamping in the inner loop of a gradient-based meta-learning algorithm leads to unstable training. Instead, we found that a multitask loss function that tracks both SDF values as well as occupancy, i.e. the sign of the signed distance function, provides stable training, increases the level of detail of the reconstructed shapes for all models, and provides faster convergence. Instead of predicting a single signed distance output, we predict two values: the distance to the surface and the sign of the distance function (inside or outside the object). The $\ell_1$ SDF loss is then combined with a binary cross-entropy sign loss with the uncertainty loss weighting scheme described by Kendall et al. [44]. This strategy can be interpreted as combining the loss terms of the concurrently proposed DeepSDF and Occupancy Networks [1, 45]. Tables 2 and 3 report mean and median Chamfer distance as well as MIOU of reconstructed meshes

Table 2: Mean / median / std Chamfer Distance (CD) for ShapeNetV2. All results are $\times 10^{-3}$.

| | Full context | | Levelset only | |
| | DeepSDF | MetaSDF | PointNet | MetaSDF Lvl. |
| --- | --- | --- | --- | --- |
| Planes | 0.07 / 0.03 / 0.21 | **0.05** / **0.02** / 0.13 | 0.55 / 0.22 / 1.07 | **0.13** / **0.07** / 0.19 |
| Tables | 0.31 / **0.04** / 1.48 | **0.13** / 0.06 / 0.24 | 0.69 / 0.21 / 1.21 | **0.32** / **0.09** / 0.95 |
| Benches | 0.31 / **0.07** / 1.70 | **0.30** / 0.09 / 1.38 | 0.74 / 0.25 / 1.45 | **0.31** / **0.11** / 0.92 |

Table 3: Mean / median / std MIOU evaluation for ShapeNetV2.

| | Full context | | Levelset only | |
| | DeepSDF | MetaSDF | PointNet | MetaSDF Lvl. |
| --- | --- | --- | --- | --- |
| Planes | 0.85 / 0.87 / 0.08 | **0.86** / **0.89** / 0.09 | 0.67 / 0.68 / 0.20 | **0.76** / **0.80** / 0.13 |
| Tables | 0.84 / 0.87 / 0.13 | **0.85** / **0.88** / 0.14 | 0.66 / 0.69 / 0.22 | **0.75** / **0.78** / 0.18 |
| Benches | **0.76** / 0.78 / 0.15 | 0.74 / **0.79** / 0.18 | 0.56 / 0.60 / 0.22 | **0.68** / **0.72** / 0.17 |

on the test set, while Fig. 8 displays corresponding reconstructed shapes. As in the 2D experiments, the proposed approach generally performs on par with the baseline auto-decoder approach. However, reconstruction of a 3D shape at test time is *significantly* faster, improving by more than one order of magnitude from 8 seconds to 0.4 seconds. To evaluate the impact of our proposed composite loss, we compare DeepSDF trained using our composite loss function against the same architecture trained using the original $\ell_1$ loss with clamping. As opposed to the proposed approach, where the composite loss is critical to stable training, DeepSDF profits little, with mean and median Chamfer Distances on ShapeNet planes at $6.7 \times 10^{-5}$ and $3.3 \times 10^{-5}$, respectively.

Next, as in our 2D experiments, we demonstrate that the same MetaSDF model—with no changes to architecture or loss function—can learn to accurately reconstruct 3D geometry conditioned only on zero-level set points, i.e., a point cloud. We compare to a baseline with a PointNet [5] encoder and conditioning via concatenation, similar to the model proposed in Occupancy Networks [45]. We match the parameter counts of both models. Tables 2 and 3 report the Chamfer distance and MIOU of test-set reconstructions for this experiment, while Fig. 9 shows qualitative results. Consistent with our results on 2D SDFs, the proposed MetaSDF approach outperforms the set-encoder based approach. Again, performance differences are most significant with out-of-distribution shapes. We note that auto-decoder-based models cannot perform zero-level set reconstructions without the aid of additional heuristics.

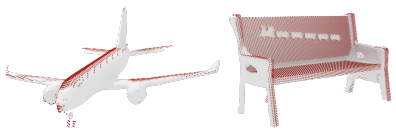

| | MetaSDF | PointNet |
| --- | --- | --- |
| Planes | **0.63** / **0.28** / 0.95 | 0.76 / **0.28** / 1.59 |
| Benches | **1.68** / 0.72 / 4.69 | 1.74 / **0.63** / 3.91 |

Figure 10: Qualitative and quantitative results of SDF reconstruction from depth maps. All quantitative results are Chamfer Distance $\times 10^{-3}$.

**Depth map completion.** We demonstrate that the prior learned by MetaSDF enables reconstruction from partial observations in the form of a depth map. We render out depth maps of ShapeNet models and transform them into world coordinates using ground-truth camera parameters. We then reconstruct the full SDF using the proposed MetaSDF model trained on level-set points, without any further fine-tuning. Figure 10 shows quantitative and qualitative results on objects from the Shapenet "benches" and "airplane" classes, compared with the PointNet-encoder based approach. We note that auto-decoder based approaches such as DeepSDF require additional heuristics to solve this problem, but may produce compelling results when these heuristics are designed appropriately.

**Visualizing inner loop steps.** To gain additional insight into the process of the inner-loop optimization, in Fig. 11, we visualize the evolution of the signed distance function over inner loop iterations, beginning with the unspecialized meta-network. The underlying model was trained for level-set

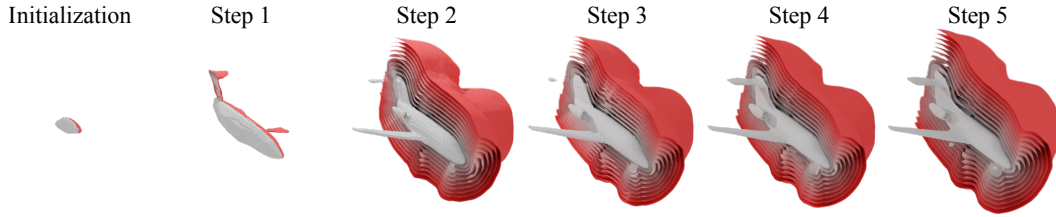

Figure 11: Evaluation of the SDF level sets throughout the inner-loop iterations. Red contours are level sets in increments of 0.05.

inference on the ShapeNet planes class. Intriguingly, even though this model is class-specific, the initialization does not resemble an airplane, and only encodes valid signed distances starting from the second iteration. At initialization and after the first update step, function values increase significantly faster than those of the ground-truth signed distance function, visualized by the collapse of the level sets onto the surface. In contrast, the mean zero-vector of the auto-decoder based DeepSDF model encodes an object that resembles an "average plane" (see supplemental material). Future investigation of this difference may reveal additional insight into how gradient-based meta learning algorithms succeed in conditioning the optimization problem for fast convergence.

**Reconstruction of objects from different class.** We evaluate the performance of models trained on the ShapeNet tables class in reconstructing objects from the ShapeNet benches class. Please see the supplemental material for quantitative and qualitative results. In contrast to our 2D results, DeepSDF outperforms MetaSDF in the case of dense conditioning, while MetaSDF continues to outperform the PointNet-encoder based approach in the case of level set conditioning. We hypothesize that in the case of dense conditioning in 3D, the unconstrained number of optimization steps of the auto-decoder based approach offers an advantage over the constant number of optimization steps of MetaSDF. This could potentially be addressed by increasing the number of inner-loop optimization steps of MetaSDF at test time.

## 5   Discussion

In summary, MetaSDF is a meta-learning approach to learning priors over the space of SDFs represented by fully connected neural networks. This approach performs on par with auto-decoder based approaches while being an order of magnitude faster at inference time, outperforms pooling-based set-encoder methods, and makes weaker assumptions about the dimensionality of the latent space. However, several limitations remain. The current implementation requires backpropagation through the unrolled inner-loop gradient descent steps, and thus the computation of second-order gradients, which is highly memory-intensive. This may be addressed by recently proposed implicit gradient methods that offer memory complexity independent of the number of inner-loop optimization steps [37]. More generally, significant progress has recently been made in gradient-based meta-learning algorithms. We leverage Meta-SGD [39] as the inner-loop optimizer in this paper, but recent work has realized significant performance gains in few-shot learning via optimization in latent spaces [36] or by learning iterative updates in an infinite-dimensional function space [46], which could further improve MetaSDFs. Another promising line of future investigation are recently proposed attention-based set encoders [33], which have been shown to perform better than set encoders that aggregate latent codes via pooling, though the memory complexity of the attention mechanism makes conditioning on large (5k points) context sets as in our 3D experiments computationally costly. Gradient-based meta-learning may also be a promising approach to learn algorithms that infer latent variables given an image in generative adversarial networks. Finally, future work may apply this approach to representations of scenes, i.e., both 3D geometry and appearance, via neural rendering [8, 22].

Our approach advances the understanding of generalization strategies for emerging neural implicit shape representations by drawing the connection to the meta-learning community. We hope that this approach inspires follow-on work on learning more powerful priors of implicit neural shape representations.

## Broader Impact

Emerging neural implicit representations are a powerful tool for representing signals, such as 3D shape and appearance. Generalizing across these neural implicit representations requires efficient approaches to learning distributions over functions. We have shown that gradient-based meta-learning approaches are one promising avenue to tackling this problem. As a result, the proposed approach may be part of the backbone of this emerging neural signal representation strategy. As a powerful representation of natural signals, such neural implicits may in the future be used for the generation and manipulation of signals, which may pose challenges similar to those posed by generative adversarial models today.

## Acknowledgments and Disclosure of Funding

We would like to offer special thanks to Julien Martel, Matthew Chan, and Trevor Chan for fruitful discussions and assistance in completing this work. Vincent Sitzmann was supported by a Stanford Graduate Fellowship. Gordon Wetzstein was supported by an NSF CAREER Award (IIS 1553333), a Sloan Fellowship, and a PECASE from the ARO.

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
