[Supplementary Material]

# MetaSDF
# –Supplementary Material–

**Vincent Sitzmann**[*]
Stanford University
sitzmann@cs.stanford.edu

**Eric R. Chan**[*]
Stanford University
erchan@stanford.edu

**Richard Tucker**
Google Research
richardt@google.com

**Noah Snavely**
Google Research
snavely@google.com

**Gordon Wetzstein**
Stanford University
gordon.wetzstein@stanford.edu

vsitzmann.github.io/metasdf/

## Contents

---

[*]These authors contributed equally to this work.

# 1 Conditioning via concatenation as a special case of Hypernetworks

Here, we demonstrate that conditioning via concatenation is a special case of a hypernetwork, where the hypernetwork is a single affine layer that only predicts the biases of the hyponetwork.

We first formalize a hypernetwork $\Phi$ that predicts the weights of a single layer of some hyponetwork. The hypernetwork maps a code vector $\mathbf{z} \in \mathbb{R}^n$ to the weights $\mathbf{W} \in \mathbb{R}^{m \times l}$ and biases $\mathbf{b} \in \mathbb{R}^l$ of the hypo-layer:

$$\Psi : \mathbb{R}^n \to \mathbb{R}^{mln}, \quad \mathbf{z} \mapsto \Psi(\mathbf{z}) = (\mathbf{W}_{(0,0)}, ..., \mathbf{W}_{(m,l)}, \mathbf{b}_0, ..., \mathbf{b}_m) \tag{1}$$

We now analyze a single layer of a neural network with conditioning via concatenation. This single layer $\mathbf{y}$ has inputs $\mathbf{x} \in \mathbb{R}^k$, $l$ hidden units, a weight matrix $\mathbf{W} \in \mathbb{R}^{l \times (k+n)}$, and bias vector $\mathbf{b} \in \mathbb{R}^l$, and is conditioned on a code vector $\mathbf{z} \in \mathbb{R}^n$ via concatenation with $\mathbf{x}$. We are only interested in the weights and biases and therefore omit the nonlinearity. This can be formalized as follows:

$$\mathbf{y} = \mathbf{W} \cdot (\mathbf{x}\|\mathbf{z}) + \mathbf{b} \tag{2}$$

where $(\cdot\|\cdot)$ signifies concatenation. We can now split up the weight matrix $\mathbf{W}$ by rows into two weight matrices, $\mathbf{W}_{\mathrm{hypo}} \in \mathbb{R}^{l \times k}$ and $\mathbf{W}_{\mathrm{hyper}} \in \mathbb{R}^{l \times n}$, and re-write $\mathbf{y}$ as follows:

$$\mathbf{y} = \mathbf{W}_{\mathrm{hypo}} \cdot \mathbf{x} + \underbrace{\mathbf{W}_{\mathrm{hyper}} \cdot \mathbf{z} + \mathbf{b}}_{\Psi_{\mathrm{bias}}(\mathbf{z})} \tag{3}$$

The affine term $\mathbf{W}_{\mathrm{hyper}} \cdot \mathbf{z} + \mathbf{b}$ does not depend on the input, only on the latent code $\mathbf{z}$, and is additive with the input-dependent linear term $\mathbf{W}_{\mathrm{hypo}} \cdot \mathbf{x}$. We can thus identify it as a "conditional bias" that is computed by a hypernetwork $\Psi_{\mathrm{bias}}(\mathbf{z}) = \mathbf{W}_{\mathrm{hyper}} \cdot \mathbf{z} + \mathbf{b}$, where the parameters $\mathbf{W}_{\mathrm{hyper}}$ and $\mathbf{b}$ are intepreted as parameters of the hypernetwork. Conditioning via concatenation is thus equivalent to a hypernetwork with a single affine layer that only predicts the biases of a hyponetwork.

# 2 2D Experiments

## 2.1 Additional Results

We provide mean and standard deviation for quantitative 2D SDF experiments for the conditional neural process and MetaSDF results in tables 1 and 2. In addition to the conditional neural process with a 4-layer ReLU set encoder, we report mean and standard deviation for a conditional neural process with a 9-layer ReLU set encoder. While the number of layers more than doubled, performance increases only slightly, still lagging far behind the performance of the proposed MetaSDF approach.

## 2.2 Reproducibility

Here, we provide exact specifications of the 2D experiments to ensure reproducibility. All code and datasets will be made publicly available.

Table 1: $\ell_1$-error (mean/std.) for reconstructions from dense and levelset observations. All results are $\times 10^{-6}$.

|  | Dense | Zero-level set |
|---|---|---|
| Cond. NP, 4-layer set encoder | 101.7/5.1 | 154.9/1.0 |
| Cond. NP, 9-layer set encoder | 92.5/2.0 | 145.8/1.2 |
| MetaSDF | **15.4/0.2** | **56.6/0.5** |

Table 2: $\ell_1$-error (mean/std.) for reconstruction of out-of-distribution samples. All results are $\times 10^{-6}$.

|  | Unseen Digit | Rotation | Composit. |
|---|---|---|---|
| Cond. NP, 4-layer set encoder | 252.4/5.0 | 301.1/10.3 | 484.9/13.6 |
| Cond. NP, 9-layer set encoder | 223.4/4.2 | 291.2/8.6 | 468.0/22.8 |
| MetaSDF | **22.8/0.5** | **33.3/1.1** | **59.8/1.2** |

### 2.2.1 Datasets

For all single-MNIST experiments, we use the MNIST dataset, as supplied with the pytorch torchvision library (link), as a starting point. We use the official train-test split of the MNIST dataset, and further split 1000 examples off the training set as a validation set. For the double-MNIST and triple-MNIST experiments, we use the datasets provided by [5] (link). The large size of the triple-MNIST test set makes it infeasible to reconstruct it using the auto-decoder framework. We thus only pick 1000 elements of the triple-MNIST dataset as a test set. To extract 2D signed distance functions, we first binarize the greyscale image, and then apply two unsigned 2D distance transforms as implemented in the scipy package, once on the original binary image and once on its inverse binary image, and combine them into a single signed distance function.

### 2.2.2 Model Details

All models are implemented as fully connected ReLU-MLPs with 256 hidden units and no normalization layers. The SDF network $\Phi$ is implemented with four layers (i.e., two hidden layers). The concatenation-based conditioning is performed by concatenating the latent code once to the input and once to the third layer of $\Phi$. The set encoder of CNPs is implemented with four layers. Hypernetworks are implemented with three layers as in [4], predicting all parameters (weights and biases) of $\Phi$. The proposed approach performs 5 inner-loop update steps, where we initialize $\alpha$ as $1 \times 10^{-1}$.

### 2.2.3 Out-of-distribution experiments

We conduct all out-of-distribution experiments using models trained on *dense* samples from the signed distance function, and condition on dense samples at test-time as well. We note that baseline approaches performed much better in the dense setting, and this thus enables the fairest comparison of the proposed approach and baseline approaches.

### 2.2.4 Hyperparameters

We train all models with ADAM with a learning rate of $1 \times 10^{-4}$ and a batch size of 32, sampling $32^2$ points for the dense SDF experiments and $512$ points for the levelset experiments. We train encoder- and meta-learning based models for 50 epochs. Due to slower convergence of the auto-decoder framework, we train auto-decoder models for 150 epochs.

### 2.2.5 Computing Hardware

We train all models on a single NVIDIA RTX 6000 GPU. Hypernetworks, conditioning via concatenation-based models, as well as conditional neural processes consume approximately 3GB of GPU memory at training time. The proposed MetaSDF approach consumes approximately 5GB of GPU memory at training time.

### 2.2.6 Runtime & Complexity

Conditional neural processes train in about 3 hours, while MetaSDF trains in about 5 hours. The training time of auto-decoder based approaches grows significantly with the size of the dataset. For out-of-distribution experiments such as the rotated MNIST digits or the compositionality experiments, the size of the datasets required 24 hours of training per model, and another 24 hours of test-time reconstruction. The memory complexity of the proposed Meta-Learning approach is $O(nm)$, with the number of query points $n$ and the number of inner gradient descent steps $m$.

### 2.2.7 Number of Evaluation Runs

Each method was run on the test set exactly three times to calculate mean and standard deviation of the final result.

## 3 3D Experiments

### 3.1 Additional Results

### 3.1.1 "Average airplane" of concatenation-based approach vs. Meta-network initialization

MetaSDF initialization    DeepSDF "mean plane"

Figure 1: Comparison of the zero-level set of MetaSDF before specialization and the "mean plane" of DeepSDF, i.e., the zero-level set of the airplane encoded by the all-zero latent code.

In Fig. 1, we show the zero-level set of the meta-initialization discovered by the proposed MetaSDF approach as well as the zero-level set of the SDF corresponding to the all-zero "mean latent code" as discovered by DeepSDF [3]. Intriguingly, they bear little resemblance. Future work may investigate the properties of the initialization of MetaSDF, potentially providing insight into why this initialization enables such swift specialization with only five gradient descent steps.

### 3.1.2 Qualitative comparison to DeepSDF and PointNet encoder on Shapenet V2 Tables

Fig. 2 and Fig. 3 show qualitative comparisons of the proposed MetaSDF approach to DeepSDF and a pointnet-encoder based approach on the Shapenet V2 tables class.

### 3.1.3 Qualitative comparison on ShapeNet V2 Benches

Fig. 4 shows a qualitative comparison of all models on ShapeNet V2 Benches.

### 3.1.4 Quantitative comparison on ShapeNet V2 Chairs

Table 3 provide Chamfer Distance and MIOU for all models on the ShapeNet V2 Chairs split. Unfortunately, we were not able to train all of our models for the full training length of our previous models, and as such, these experiments are not consistent with our prior experiments. We will rerun the experiment on ShapeNet V2 Chairs and add the results to the main paper when computational resources become available.

### 3.1.5 Quantitative comparison for models trained on ShapeNet V2 Tables and evaluated on ShapeNet V2 Benches

Fig. 5 provides a qualitative comparison and Table 4 provides results for all models trained on ShapeNet V2 Tables and evaluated on ShapeNet V2 Benches.

Figure 2: Qualitative comparison of the proposed MetaSDF approach and DeepSDF on the Shapenet v2 tables dataset.

Figure 3: Qualitative comparison of the proposed MetaSDF approach and a PointNet Encoder on zero-level set reconstructions from the Shapenet v2 tables dataset.

### 3.1.6 Inference-time convergence plot

In order to better illustrate an advantage of MetaSDF over auto-decoder methods that achieve similar reconstruction accuracy, Fig. 6 plots reconstruction error, the percentage of sample points misclassified as inside or outside the object, when fitting an object at inference-time. While MetaSDF converges in a constant five steps, auto-decoder methods require optimizing a latent code over many hundreds of gradient descent steps and are thus an order of magnitude slower when fitting new objects at inference time.

Table 3: Preliminary Mean / median / std Chamfer Distance (CD) and MIOU for ShapeNetV2 Chairs. CD results are $\times 10^{-3}$.

|  | Full context | | Levelset only | |
|---|---|---|---|---|
|  | DeepSDF | MetaSDF | PointNet | MetaSDF Lvl. |
| CD | **0.17** / **0.07** / .278 | 0.26 / 0.09 / 1.10 | 1.00 / 0.68 / 1.03 | **0.28** / **0.18** / 0.33 |
| IOU | 0.82 / 0.87 / 0.16 | **0.85** / **0.90** / 0.15 | 0.61 / 0.63 / 0.22 | **0.77** / **0.81** / 0.18 |

Table 4: Mean / median / std Chamfer Distance (CD) and MIOU for models trained on ShapeNetV2 Tables and evaluated on ShapeNet V2 Benches. CD results are $\times 10^{-3}$.

|  | Full context | | Levelset only | |
|---|---|---|---|---|
|  | DeepSDF | MetaSDF | PointNet | MetaSDF Lvl. |
| CD | **0.55** / **0.17** / 2.06 | 1.34 / 0.40 / 5.16 | 2.67 / 2.12 / 3.38 | **1.48** / **0.95** / 1.60 |
| IOU | **0.56** / **0.59** / 0.18 | 0.51 / 0.52 / 0.23 | 0.28 / 0.20 / 0.22 | **0.33** / **0.35** / 0.21 |

Figure 4: Qualitative comparison of models on the ShapeNet V2 Benches class.

Figure 5: Qualitative comparison of models trained on ShapeNet V2 Tables and evaluated on ShapeNet V2 Benches.

## 3.2 Reproducibility.

Here, we provide exact specifications of the 3D experiments to ensure reproducibility. All code and datasets will be made publicly available.

### 3.2.1 Datasets.

We utilize the 'Planes' and 'Tables' categories of ShapeNetV2 [1] (link) for all 3D experiments. We preprocess SDF samples, used in dense reconstruction, and surface samples, used in zero-level set reconstruction, with the preprocessing code (link) from DeepSDF [3]. We use DeepSDF's train-test splits for both object categories, but extract 20 examples from each training set for use as validation sets. We encountered errors in using DeepSDF's preprocessing code and thus were only able to process about 80% of the 3D models.

Figure 6: Reconstruction error against iterations for MetaSDF and DeepSDF at test-time.

### 3.2.2 Model Details.

**Details on composite loss**  Empirically, we found that a composite loss function that combines the $\ell_1$ loss on the value of the predicted SDF with a binary crossentropy loss on the sign of the predicted SDF lead to better results for MetaSDF. Because meshes are represented by the zero-level set, precisely predicting the sign of the SDF close to the surface is far more important to mesh accuracy than accurately predicting the value of the SDF away from the surface. Intuitively, our strategy encourages the model to emphasize not only the value but also the sign of SDF predictions. Our model predicts two outputs: a prediction of signed distance and a sigmoid output representing the sign. At training time, we calculate the $\ell_1$ loss on the SDF prediction and the binary crossentropy loss on the sign prediction. We combine these loss terms with the method proposed by Kendall et al. [2]. At test time, we combine the two outputs by multiplying the absolute value of the distance prediction with the predicted sign to produce a single SDF prediction.

**Dense MetaSDF**  MetaSDF for our dense experiments is implemented as a fully connected ReLU-MLP with 8 hidden layers of 512 units and no normalization layers. We utilize 5 inner-loop update steps and per-parameter, per-step learning rates, which are initialized to $5 \times 10^{-3}$.

**Zero-Level set MetaSDF**  We utilize a similar architecture for our zero-level set experiments, but in order to better match parameter counts to PointNet encoders, we reduced network size to 5 hidden layers of 512 units.

**DeepSDF**  We utilize the official DeepSDF implementation, which consists of 8 hidden layers of 512 units and weight normalization. DeepSDF uses an autodecoder, with latent codes of 256 units.

**PointNet Encoder**  We implement a resnet PointNet encoder, which consists of 5 resnet blocks of 512 units and encodes a 256-dimensional latent code. The SDF network, $\Phi$ , is implemented with 5 hidden layers of 512 units.

### 3.2.3 Hyperparameters.

We train all models with ADAM and initialize learning rates to $5 \times 10^{-4}$. All models were trained to convergence. We trained Dense MetaSDF for 500 epochs using our composite loss function and decayed learning rate by a factor of 0.5 after 350 steps. All other models, including Zero-level set MetaSDF, were trained using L1 loss for 2000 epochs, decaying learning rate by a factor of 0.5 every 500 steps. All models use a batch size of 64 scenes.

### 3.2.4 Computing Hardware

We train all models on dual NVIDIA RTX 8000 GPUs.

### 3.2.5 Runtime & Complexity.

The complexity of the proposed Meta-Learning approach is $O(nm)$, with the number of query points $n$ and the number of inner gradient descent steps $m$. DeepSDF and PointNet Encoders train to 2000 epochs in approximately 4 days while MetaSDF trains to 2000 epochs in approximately 6 days.

### 3.2.6 Number of Evaluation Runs.

Because we randomly sample context points during mesh reconstruction, reconstructed meshes vary between runs. Following the evaluation procedure of DeepSDF, at test-time each model reconstructs each mesh twice; the better reconstruction is used in computing chamfer distance across the test set. This evaluation procedure was conducted once for each model.