[Reviews · NeurIPS 2020]

Review 1

Summary and Contributions: This paper proposes using gradient-based meta-learning to learn a space shapes encoded which are represented as neural signed distance functions. Existing techniques for shape generation are based on either set-based auto-encoders which have been shown to underfit for large inputs, or auto-decoders which require slow inference time optimization. In contrast, the proposed technique shows generalization performance on par with auto-decoder based methods while being an order of magnitude faster: requiring only a small constant number of inference time optimization steps.

Strengths: This is a novel approach to the problem of conditional shape generation, offering many potential benefits over existing techniques. I think that the meta-learning formulation is nice way of thinking about learning shape priors and is more amenable to tuning for downstream tasks than traditional techniques based on latent codes. Furthermore, I think the results of the paper were promising, demonstrating how the proposed technique overcomes many of the limitations of previous methods.

Weaknesses: While the ideas in this paper are novel and the results promising, I have several concerns that I feel the authors should address. In the reconstruction from surface points only, auto-decoder methods such as DeepSDF naturally fail since the predicted function quickly degenerates to zero. That said, in downstream shape completion tasks, one almost certainly has access to some estimate for the surface normal, in which case auto-decoders can be used. (Note that if normals are not returned by the sensor, these can almost trivially be computed from depth maps using off the shelf algorithms). I would like to see some kind of experiment evaluating partial shape reconstruction with surface information (see e.g. Figure 8 in DeepSDF). I think the partial shape completion experiment is important for two reasons: The first is that it shows MetaSDF in a real-world downstream task, something that is lacking in the current version of the paper. The second reason is that all the current experiments appear to use samples which are either dense in the volume or dense on the surface. It's thus unclear to me how MetaSDF will generalize with only partial information, and how this generalization compares to other state of the art methods. I also feel that the use of only two ShapeNet classes may not be enough. The paper compares against DeepSDF and Occupancy Networks which show their results on 5 and 13 ShapeNet classes respectively. I think this is important to show how MetaSDF adapts to a variety of shapes. At the very minimum, I would like to see this model evaluated on a class such as benches which contains models with fine topological features. Capturing small shape details is a very important component of learning to represent 3D objects and I think it's important to spell out explicitly how MetaSDF compares to the state of the art in this regard. Finally, in addition to reporting the Chamfer distance in experiments, I think the paper should show the Intersection-over-Union (IoU) for their shapes. IoU is a standard metric for benchmarks in shape reconstruction. I think this metric is particularly relevant for volumetric reconstruction techniques. For signed distance functions, IoU can easily be approximated by scattering many points in the volume and evaluating their occupancy.

Correctness: The claims in the paper appear correct to me. I think the methodology is correct for the evaluations that the authors performed, but I feel that some experiments are lacking.

Clarity: I found the paper to be very clear and well written.

Relation to Prior Work: Yes. This is quite explicit.

Reproducibility: Yes

Additional Feedback: I really want to accept this paper based on its core concepts, but I feel there are some missing experiments. If the authors can justify why these are not needed or provide them, I would happily increase my score. After reviewing the author feedback, I feel like their experiments and comments justify an accept for MetaSDF. I would still like to see a complete evaluation on ShapeNet for the final revision.


Review 2

Summary and Contributions: The paper proposes to use meta-learning principles for learning implicit shape representations in order to better generalize across different shapes and to speed up test time inference. In contrast to the established auto-decoder framework the proposed approach is an order of magnitude faster at test time. The key idea is leverage training mechanisms from model-agnostic meta learning (MAML) to learn a general shape representation that can then be quickly refined within very few gradient steps to specialize for a particular shape. This drastically reduces the amount of optimization steps needed to compute a shape representation at test time.

Strengths: + The paper generalizes a recent approach for learning implicit shape representations leading to state-of-the art shape reconstruction results at significantly lower test time results. + The paper provides interesting insights about the learning of neural implicit shape representations and their generalization.

Weaknesses: - While the paper aims at better generalizability across shapes, the experiments are still limited to generalize only within a shape category (like planes or cars). The authors find that the learned generalized shape does not resemble the average shape of the learned category (L267). This indicates that generalization might actually also work across shape categories which is certainly a more ambitious goal, but given the motivations of the paper this is also something that could have been investigated. - Experimental evaluation: Given the large amount of related work in this area, the SOTA comparison could be more extensive and additional comparisons to IM-NET, DISN would better support the SOTA claims of the paper.

Correctness: The proposed methodology is sound and the drawn conclusions seem very reasonable.

Clarity: The paper is very well written and structured. The figures support the text well.

Relation to Prior Work: The paper covers the related work well. Though, there are many recent related papers which were not yet published at the submission deadline, but available on arxiv. One published paper that could be added is: [A] Qiangeng Xu, Weiyue Wang, Duygu Ceylan, Radomir Mech, Ulrich Neumann, DISN: Deep Implicit Surface Network for High-quality Single-view 3D Reconstruction, NeurIPS 2019

Reproducibility: Yes

Additional Feedback: - L88: States: “We may alternatively choose to approximate the binary ‘occupancy’ of points rather than their signed 89 distances”. This sounds you are doing either one the two, but I guess this sentence refers to L239ff which states that two losses are combined for fast convergence. Or did you also try learning occupancies alone? - L161/L43: it would be good to add references to MNIST and ShapeNet on the first occurrence in the paper. - Tables 1, 2, 3: I don’t see the purpose of rescaling all numbers with 10^-3. Wouldn’t it make more sense to get rid of the “0.” in front of all numbers since having less (useless) digits in the table makes it more readable?


Review 3

Summary and Contributions: This paper proposes to formulate the learning of neural implicit representations of shapes as a meta-learning task. This optimally conditions the weights of the implicit shape network so that at inference time a new shape can be fitted using only a few gradient steps. The authors claim that this approach is significantly faster than auto-decoder approaches, allows for a variable number of observations, and outperforms polling-based encoders in this regard. These claims are well validated via the experimental results.

Strengths: The main strength of the paper is that it is extremely well written and clearly presented. The contributions are well framed against existing work (auto-decoders, set-encoders, meta-networks) and the advantages of the approach are made very clear. For people in the field, the idea itself is easy to understand and should be easy to implement and reproduce. Furthermore, the motivation behind the approach (i.e. a fast approach for inferring shapes from multi-view observations) is also very convincing and perfectly pitched to the reader. The experimental results are convincing and support the claims made about speed and accuracy. The approach is tested both on the 2D MNIST dataset as well as the 3D shapenet dataset. I would, however, have liked to see more in terms of the speed analysis as this is what the authors claim to be the main advantage of the approach (see weaknesses).

Weaknesses: The numerical comparisons on Shapenet are only given for the “planes” and “tables” class. While this is not a critical weakness, it would be good to know whether a similar trend is observed for the other classes as well? The approach requires second-order gradients during training, it is quite memory intensive but the actual complexity is not given. Also, how does this compare, relatively, to the auto-decoder approach? As the main advantage is computation time, and both the auto-decoder and the proposed method are iterative approaches, it would be good to show the error as a function of the iterations. This would make it easier to see the difference in the convergence between these two approaches. Also, it would be good if the authors specified the convergence criteria, e.g. in Section 4.1 it states that the auto-decoder and meta-networks take 4s to converge while the proposed method takes 50ms, but what determines this convergence? GANs also have a difficulty with inferring the latent representation corresponding to particular inputs as they don’t have an encoder. Would this approach work well for learning an inference module for GANs as well? A more general question as well is what makes this approach particularly suited for inferring shape representations? The technical novelty of the approach is somewhat limited. It is a straightforward application of MAML to the context of implicit shape representation networks. What are the technical challenges that the authors encountered when designing the approach and what was done to overcome them? Or was it just a straightforward application of MAML? I think it would be good if the authors could clarify the technical differences and similarities compared to MAML.

Correctness: In general, the approach, description and empirical methodology appears to be technically sound. The experiments are well designed allow for the verification of the claims regarding accuracy. It would maybe be good to report the standard deviation of the Chamfer distance in Table 2 and 3 so the reader can get a sense of the distribution of errors across the shapes. I also think the paper is currently missing a thorough evaluation of the runtime as only one number (50ms) is reported and no convergence criteria are given.

Clarity: As mentioned above, the paper is very well written and the authors need to be commended for the effort put into the writing. There is really nothing to fault in terms of the exposition of the approach. The only part that I found a bit unclear is the description of the auto-decoder as a meta-learning approach. While I can see what the authors are alluding to, I’m not sure what is meant by “[the auto-decoder] does not perform model specialization in the forward pass”.

Relation to Prior Work: The related work covers all the relevant approaches and in general the proposed approach is clearly framed with respect to the existing literature. I think the only improvement that could be made is to clarify the differences and similarities relating to the use of MAML in the approach. If no changes are made, I think the authors should clearly state this to avoid confusion.

Reproducibility: Yes

Additional Feedback: There are only some very minor typos: ln 69: only few -> only a few ln 73: via few steps -> via a few steps ########## Post-rebuttal ############# After the rebuttal, other reviews and discussion I will keep my current score. The authors have addressed most of my main concerns and I think it is a very well presented paper, describing an interesting idea that is supported by good results. My only critique remains that the novelty is somewhat limited in that it is basically using MAML instead of standard SGD for refining the representation. That said, the authors have pointed out some interesting challenges they had applying MAML to this problem in the rebuttal which I hope they will add to the paper.


Review 4

Summary and Contributions: The paper formulate learning of a shape space as a meta-learning problem and leverage gradient-based meta-learning algorithms. We demonstrate that this approach performs on par with auto-decoder based approaches while being an order of magnitude faster at test-time inference. I agree with most of the concerns of the other reiviews, and the author has properly addressed some of them in the rebuttal. Experiments e.g. partial point completion, cross category testing should be added, and the technical challenges to apply MAML should be made clear in the next draft. Given those, I am would raise the rating to 6.

Strengths: It seems to have some nice results, but the paper is confusing to read, thus I personally does not have a clear picture about the goal it tries to achieve. I hope the author could clear my confusions in below, and in future draft, include figure illustrations, as well as clearer description of the test time behavior. Some of my understanding

Weaknesses: The problem settings of few-shot learning as well as meta learning is often confusing to people not familiar with the concept, extra illustration is needed for easier appreciation of the work. My guess for the task of this paper is, in test time, you are given some partial observation, and you want to quickly fit a function by doing limited number of gradient descent. To achieve this, you want to have a good itialization ( \theta?) and appropriate learning rate \alpha. To get those, you employ algorithm 1 . If that is indeed the task, there's some confusing pieces of algorithm 1, which I hope the author could kind clarify for me: what does l_1 mean in Ln.133? It seems the parameters of the network to optimize in test time differs for different network architecture? only latent code to optimize for auto-decoders (Ln.112-114)? what parameters are subject to optimize for hypernetworks? What are the settings for "MetaSDF" in experiments? Is Ln.174 the said MetaSDF network architecture? I had a feeling that my confusion may be caused by the fact that Ln.93-115 should be placed inside related works, rather than in the main approach.

Correctness: see above

Clarity: As above, I personally feel the paper difficult to understand.

Relation to Prior Work: yes

Reproducibility: Yes

Additional Feedback:

[Author Response · NeurIPS 2020]



Meta Deep GT (a) — Meta Point GT (b) — (c) — (d) — (e)

We are glad that the reviewers found **MetaSDF** to be a "novel approach [with] many potential benefits" **(R1)**, the
"motivation [...] very convincing and perfectly pitched to the reader" **(R3)**, and "providing interesting insights about
the learning of neural implicit shape representations" **(R2)**. Neural implicit representations are an emerging field
with applications across vision, graphics, and other branches of ML. We propose a novel approach to generalization
across these emerging representations, with clear benefits on inference time and out-of-distribution generalization
(Sec. 4). We stress that we *do not* claim a new state-of-the-art approach to shape reconstruction **(R2)**, merely that the
proposed approach performs *on par* with state-of-the-art approaches (see ln. 47, ln. 246). Instead, our key contribution
is establishing the connection between the emerging fields of neural implicit representations and gradient-based meta-
learning. This is the very first step, and a wealth of existing work on meta-learning may provide significant further
improvements. We believe that this will spur follow-up work benefitting both of these promising research directions.

**Further ShapeNet classes (R1, 3)** We have trained models on the ShapeNet "benches" class—please see qualitative
test-time reconstructions for dense SDF samples (a), and surface samples only (b). Quantitatively, we achieve a Chamfer
Distance (CD) of mean $3.01\mathrm{e}{-4}$, median $8.54\mathrm{e}{-5}$, stddev $1.37\mathrm{e}{-3}$ (ours) vs. mean $3.11\mathrm{e}{-4}$, median $7.36\mathrm{e}{-5}$, stddev
$1.7\mathrm{e}{-3}$ (DeepSDF) while outperforming the PointNet-encoder model. We will benchmark another class with fine
features for the final manuscript. We have done our best to benchmark the proposed approach extensively, but note that
each class takes several days to train on two of the largest available NVIDIA GPUs, the 48 GB RTX8000, which we
share with other researchers. We note that 2D results (Sec. 4) are entirely consistent with our 3D results. Together, this
conclusively demonstrates the potential of MetaSDF. Research on gradient-based meta-learning is progressing quickly,
and it is poised to become more computationally affordable in the near future (see ln. 283).

**Reconstruction from partial observations (R1)** We ran an experiment to reconstruct SDFs from depth maps as in
Fig. 8 of DeepSDF—see qualitative result in (c)—with no further fine-tuning or heuristics. Quantitatively, we achieve a
Chamfer Distance of $6.55\mathrm{e}{-4}$ for planes, where DeepSDF reports $1.16\mathrm{e}{-3}$. We use the results supplied by the authors
of DeepSDF, as the authors do not provide code or data for this experiment and we did not succeed in reproducing their
results. We will add experiments and comparisons with further classes to the final manuscript.

**Related work & IM-NET (R2)** We will discuss DISN in-depth. IM-NET reconstructs shapes using either an image
encoder, which is of different modality than the 3D samples MetaSDF is focused on, or a PointNet encoder with
conditioning via concatenation. We benchmark against this architecture (see submission Table 3, Fig. 8).

**MIOU (R1)** We have computed MIOU for planes, tables: $0.87$, stddev $9.26\mathrm{e}{-2}$ and $0.85$, stddev $1.43\mathrm{e}{-1}$ (MetaSDF)
vs. $0.85$, stddev $8.37\mathrm{e}{-2}$ and $0.85$, stddev $1.30\mathrm{e}{-1}$ (DeepSDF). We will add MIOU for all classes to the paper.

**Out-of-Distribution generalization (R2)** We have reconstructed benches with the model trained on tables. Please find
qualitative results in (d). We will add a quantitative benchmark to the final manuscript.

**Standard deviations (R3)** We have computed the CD std. dev. on planes and tables: $1.29\mathrm{e}{-4}$ and $2.39\mathrm{e}{-4}$ (MetaSDF)
vs. $2.07\mathrm{e}{-4}$ and $1.48\mathrm{e}{-3}$ (DeepSDF). We will include further classes and MIOU std. dev. in the final paper.

**Convergence comparison (R3)** Please see (e) for a plot of test-time iterations (log scale) vs. test loss. We define
convergence as the number of optimization steps until the test-time loss does not decrease further.

**Memory Complexity (R3)** MAML requires the computation of second-order gradients, whose memory complexity
depend on the auto-differentiation algorithm, which are evolving rapidly (see Bettencourt et al., NeurIPS 2019
workshops). We can state that MAML's memory complexity scales linearly with inner-loop gradient-descent iterations.
With the architectures described in the paper, MAML requires about twice the memory of the auto-decoder.

**Novelty in light of MAML (R3)** We do not claim any fundamental contributions to gradient-based meta-learning itself,
and will highlight this in the paper. However, we found several technical challenges specific to applying gradient-based
meta-learning to generalization across implicit neural representations. First, vanilla MAML underfits the data, and it
is necessary to leverage per-parameter, per-step tunable learning rates as proposed by Li et al. (2019), Antoniou et
al. (2018). Gradient warping Flennerhag et al. (2019) and related methods are likely to improve performance further.
Next, performing gradient descent with truncated SDF values as proposed in DeepSDF leads to unstable training &
catastrophic failure that the model does not recover from. This was addressed by the proposed composite loss function.

**Applications to GANs (R3)** Applications to GANs are an exciting avenue of future work which we will discuss.

**Improve exposition (R4)** The motivation for our work is indeed to find a good initialization that allows fitting an
implicit neural representation in few gradient descent steps. We will include an overview figure to better motivate our
proposed meta-learning approach. $\ell_1$ in ln. 133 refers to the L1 loss. A discussion of hypernetworks can be found in
"Scene Representation Networks", Sitzmann et al. (2019). Ln. 174 indeed describes the proposed architecture.

[Meta-Review · NeurIPS 2020]

This paper proposes to use a meta-learning tool for learning implicit shape representation to o better generalize across different shapes. Authors did a good job in the rebuttal which well answered most of reviewers’ concerns, leading that two reviewers raised their scores. For the final version, I would like to suggest authors to include experiments for ShapeNet things.